# VIDAR-Based Road-Surface-Pothole-Detection Method

**DOI:** 10.3390/s23177468

**Published:** 2023-08-28

**Authors:** Yi Xu, Teng Sun, Shaohong Ding, Jinxin Yu, Xiangcun Kong, Juan Ni, Shuyue Shi

**Affiliations:** 1School of Transportation and Vehicle Engineering, Shandong University of Technology, Zibo 255000, China; 21502050238@stumail.sdut.edu.cn (T.S.); 21402030147@stumail.sdut.edu.cn (S.D.); 21502050244@stumail.sdut.edu.cn (J.Y.); 22502060006@stumail.sdut.edu.cn (X.K.); 22502060003@stumail.sdut.edu.cn (J.N.); 22502060007@stumail.sdut.edu.cn (S.S.); 2Collaborative Innovation Center of New Energy Automotive, Shandong University of Technology, Zibo 255000, China

**Keywords:** road-surface pothole detection, VIDAR, monocular vision, MSER, depth update

## Abstract

This paper presents a VIDAR (a Vision-IMU based detection and ranging method)-based approach to road-surface pothole detection. Most potholes on the road surface are caused by the further erosion of cracks in the road surface, and tires, wheels and bearings of vehicles are damaged to some extent as they pass through the potholes. To ensure the safety and stability of vehicle driving, we propose a VIDAR-based pothole-detection method. The method combines vision with IMU to filter, mark and frame potholes on flat pavements using MSER to calculate the width, length and depth of potholes. By comparing it with the classical method and using the confusion matrix to judge the correctness, recall and accuracy of the method proposed in this paper, it is verified that the method proposed in this paper can improve the accuracy of monocular vision in detecting potholes in road surfaces.

## 1. Introduction

Road cracks and road-surface potholes are the most common types of road damage. Road cracks are mainly caused by substandard construction techniques and the overloading of vehicles, while road cracks evolve into road-surface potholes after being washed away by rain, struck by hard objects and crushed by vehicles [1]. Road-surface potholes not only reduce the service life of the road, but because they are depressed downwards, there is no three-dimensional height information of a conventional obstacle, which makes them difficult to identify accurately with conventional vision-detection methods. When a vehicle drives over a pothole in the road, the pothole can cause damage to the vehicle’s suspension and tires, which in turn can affect the vehicle’s performance and make it vulnerable to accidents when braking or changing lanes.

The current common method of road pothole detection relies on technicians to carry out an analytical review of road images and video data collected by road surveillance vehicles, a process that is time-consuming and susceptible to the subjective intentions of inspectors [2]. To address these drawbacks, researchers have progressively used intelligent algorithms for road-surface pothole detection. The existing automatic pothole-detection methods can be broadly classified into three methods: vision-based methods, vibration-based methods and 3D reconstruction-based methods.

The vision-based automatic pothole-detection method is the most cost-effective and common method available. It solves the road pothole-detection problem using image classification, object recognition or semantic segmentation algorithms [3]. The vibration-based automatic pothole-detection method uses the accelerometer of a smartphone to implement the detection function. However, its biggest drawback is that it requires a vehicle carrying a vibration sensor to pass through the pothole in order to complete the detection, which still causes damage to the vehicle [4]. The method based on 3D reconstruction is achieved by means of a laser scanner, which is the most accurate method of inspection and the most expensive of the three methods [5].

For laser scanning-based 3D road-surface reconstruction methods, She et al. proposed that the optimal spacing between two adjacent transverse profiles should be fully considered when reconstructing 3D models using laser technology, while verifying that a spacing of 2 mm is the optimal spacing for reconstructing 3D models [6]. Feng et al. compared five crack-detection algorithms using terrestrial laser scanner (TLS) point clouds and found that the along-range and cross-range profile-based filtering methods performed the best in crack detection [7].

For machine learning methods based on visual sensing, Egaji et al. preprocessed the data with a 2 s non-overlapping window and compared five binary classification machine learning models (plain Bayesian, logistic regression, SVM, KNN and random forest tree) and found that random forest had better processing results [8]. Lee et al. considered the minimum temperature, relative humidity, precipitation and traffic volume to build a deep learning model based on convolutional neural networks to detect potholes and calculate damage ratios (pixel method), a method that could provide the government with effective budget management and reduce road accidents caused by potholes on certain road sections [9]. Saisree et al. used tensor flow to train and test keras pothole images and found that the InseptionResNetV2 model and VGG19 model had a better detection accuracy than the Resnet50 model [10]. Especially, Bhatia et al. achieved an accuracy of 97.08% for pothole detection using a pothole-detection system based on a CNN-based ResNet model combined with thermal imaging [11].

For the vibration approach using accelerometer sensors, Lee et al. developed a system of smartphone-based dual-acquisition methods that can capture images of pavement anomalies and measure vehicle acceleration when detected, exploring the complementary advantages of the two different approaches [12]. Setiawan et al. proposed an improved U-Net architecture with an integrated bi-directional long short-term storage layer for the semantic segmentation of smartphone motion sensor data for pavement classification [13]. Notably, Pandey et al. proposed a new method for applying accelerometer data to convolutional neural networks, which is more advantageous in terms of detection accuracy and computational complexity [14].

In addition to the above methods, Li et al. fused genetic algorithms with visual sensing to achieve the 3D reconstruction and extraction of the morphological features of potholes through point cloud processing [15]. Notably, Zhang et al. used an unmanned aerial vehicle (UAV) road-damage database and described a multi-level attention mechanism called multi-level attention block (MLAB) to enhance the use of basic features using YOLO v3 (You Only Look Once version 3) [16]. However, the method is only 68.75% accurate and uses UAVs, which cannot be used directly on smart vehicles.

Particularly, R Fan et al. distinguished whether a road was damaged or not by transforming a dense parallax map, estimating the transformation parameters using golden section search and dynamic programming, then extracting undamaged road areas from the transformed parallax map using Otsu’s thresholding method, and finally reducing the effect of outliers by the consistency of random samples to achieve the fusion of 2D road images and 3D pavement modelling under binocular vision, in which the successful detection accuracy of the system was about 98.7% and the overall pixel-level accuracy was about 99.6% [17]. But this method has some limitations in measuring potholes: its setup of pothole-detection parameters cannot be applied to all situations and it cannot measure the dimensional information of potholes.

In addition, the EU-autonomous HERON project will develop an integrated automation system to design an autonomous ground-based robotic vehicle supported by autonomous drones to adequately maintain road infrastructure. The vehicle will use sensors and scanners for 3D mapping, in addition to an AI toolkit, to help coordinate road maintenance and upgrading workflows with the aim of reducing accidents, lowering maintenance costs and increasing the capacity and efficiency of the road network [18].

VIDAR [19] is a method that uses a monocular camera and an inertial measurement unit as the basic sensing unit. In order to ensure that road potholes can be effectively detected in real time while the vehicle is in motion, and to reduce the degree of damage to vehicle performance caused by road potholes, as well as the occurrence of traffic accidents, this paper proposes a VIDAR-based road-pothole-detection method. The identification process of this method is as follows: first, the rectangular frame of the road pothole is determined by extracting feature points, the width of the road pothole is calculated using the transformation relationship between pixel coordinates and world coordinates, and then the depth of the road pothole is calculated by tracking the feature points of the road pothole and using the transformation relationship between pixel coordinates and world coordinates. It is assumed that when the actual inspection is carried out, the vehicle remains in a straight line on a flat road and that the road pothole is really present, with a certain depth, width and length.

The VIDAR-based pothole-detection method for the road surface proposed in this paper has a faster processing time than the Adaboost cascade detection method. The combination of an MSER-based fast image region-matching method and the VIDAR-based obstacle detection method can bypass obstacle classification and reduce the time and space complexity of road-environment perception [20]. Meanwhile, the method fuses monocular vision with IMU to calculate the dimensional information (width, length and depth) of potholes on the road surface using camera imaging principles. The accuracy, recall and precision of the confusion matrix are compared and analyzed with the traditional method, which verifies that the method proposed in this paper has a higher accuracy.

The remainder of this paper is as follows: Section 2 describes the principles of VIDAR (a vision IMU-based detection and ranging method) for obstacle detection. Section 3 details the VIDAR-based road-surface-pothole-detection method and the obstacle feature point-tracking method, and designs the relevant calculation methods for road-surface pothole width, length and depth as well as the update method for pothole depth. Section 4 presents simulation experiments and real-vehicle tests of the proposed method and comparison experiments with conventional methods to verify its detection accuracy and speed. Section 5 provides a summary of the proposed methodology and experimental results.

## 2. The Principle of VIDAR Detection

The research in this paper is based on the VIDAR detection method for stereo obstacles. VIDAR based on monocular vision has the advantages of a simple process, easy operation and high detection accuracy, which can accurately obtain the position information of the target obstacle ahead in practical detection, and at the same time can detect unknown obstacles that cannot be identified using machine vision, which is a practical and effective obstacle-detection algorithm and the basis for the subsequent research in this paper.

In the actual obstacle-detection process, VIDAR first uses the MSER (maximally stable extremal regions)-based fast image region-matching method to extract feature points of the obstacle, and image matching is performed on two consecutive frames of the acquired image. Then, the non-obstacle points extracted using the MSER image area-matching method are eliminated by the VIDAR discrimination principle for stereo obstacles, and the obstacles in the detected image are quickly and directly identified.

### 2.1. Static Obstacle Distance Detection

As shown in Figure 1, image acquisition is the process of mapping objects in a 3D space to a 2D image plane, and this process can be simplified as a pinhole camera model.

Let us consider the lowest point P of the MSER connected to the measured area as the intersection of the obstacle and the road plane. For the sake of calculation, assume that the camera optical axis is exactly pointing at a point, as shown in Figure 1. Assume that the effective focal length of the camera is f, the distance from the optical axis of the lens to the ground is h, the pixel size is μ, the pitch angle is θ, and the coordinates of the origin of the image coordinate system is $\left(x_{0},y_{0}\right)$. It is known that the coordinates of the intersection point P of the obstacle in front of the self-vehicle and the road plane in the imaging plane is $\left(x,y\right)$. The horizontal distance between the point P and the self-vehicle camera can be expressed as
(1)$$d=\frac{h}{\tan\left(\partial +\arctan\left[\left(y_{0}-y\right)\mu /f\right]\right)}$$

Suppose point A is the imaging point of the obstacle on the $y_{1}$-axis in the previous frame, and point B is the imaging point of the obstacle on the $y_{2}$-axis in the next frame, as shown in Figure 2. Point $A^{\prime }$ is the point on the road plane corresponding to A. Point $B^{\prime }$ is the point on the road plane corresponding to B. $d_{1}$ is the horizontal distance from the self-vehicle camera at the previous frame position to the point $A^{\prime }$, $d_{2}$ is the horizontal distance from the self-vehicle camera at the next frame position to the point $B^{\prime }$, where $d_{1}$ and $d_{2}$ can be obtained by Equation (1). Δd is the distance the camera moved between the two frames before and after the shot (it is also the distance the car moved since the time period), $d_{1}=d_{2}+\Delta d$. In the actual detection, the obstacle is three-dimensional, so $d_{2}=d_{1}+\Delta d+\Delta l$. If $d_{2}\neq d_{1}+\Delta d$, then $A^{\prime }$ and $B^{\prime }$ have height, on which, if Δd is known, then static obstacles can be identified by Δl.

### 2.2. Dynamic Obstacle Distance Detection

When the obstacle in front of the self-vehicle moves in the horizontal direction (as shown in Figure 3), the horizontal distance between the previous moment of the camera and the vertex of the obstacle is $s_{1}$, and the horizontal distance between the camera and the vertex of the obstacle at the next moment is $s_{2}$, the specific relationships between $d_{1}$, $d_{2}$, $s_{1}$, $s_{2}$, and Δd are
(2)$$\left\{\begin{array}{c} d_{2}=d_{1}+\Delta l-\Delta d\\ s_{2}=s_{1}+s-\Delta d \end{array}\right.$$

According to the characteristics of the relationship between the sides and angles of a right triangle, the specific relationships between $h_{v}$, h, $s_{1}$, $s_{2}$, $d_{1}$, and $d_{2}$ are
(3)$$\left\{\begin{array}{c} \frac{h_{v}}{h}=\frac{d_{1}-s_{1}}{d_{1}}\\ \frac{h_{v}}{h}=\frac{d_{2}-s_{2}}{d_{2}} \end{array}\right.$$

$\Delta l=\left(h\times s-h_{v}\times \Delta d\right)/\left(h_{v}-h\right)$ can be obtained according to Equations (1) and (2). Thus, by tracking and calculating the positions of feature points, obstacles can be identified in any case where $h\times s\neq h_{v}\times \Delta d$.

## 3. VIDAR-Based Road-Surface-Pothole-Detection Method

The overall objective of this research is to validate the effectiveness of the proposed VIDAR-based pothole-detection method, then to verify that this method can identify potholes and calculate their sizes in real time by means of camera equipment.

Although road-surface potholes have no height information and are depressed downwards, they are detected in much the same way as real obstacles. When detecting potholes with VIDAR, the fast image region-matching method based on MSER is used to match feature points between two images obtained in a relatively short time and to eliminate redundant feature points.

In image processing, the MESR-based fast image region-matching method firstly passes the MSER extraction algorithm and fits all the MSERs (maximum extreme stable region) in the reference image and the target image into an elliptical region to provide more useful information, and then in improves the matching accuracy through the feature point detection method. Finally, the stability of MSERs is used to ignore the differences in the position and shape of MSERs in the two images, simplifying the matching process and increasing the matching speed.

Therefore, after generating the MSERs, the MSERs are marked with rectangular boxes, while the feature points at the widest and longest part of the rectangular box are extracted (as shown in Figure 4).

### 3.1. Calculation of the Width of Road-Surface Potholes

For the calculation of the width of road-surface potholes as shown in Figure 5, let A be the first imaging point of the width of the rectangular frame outside the road-surface pothole: its coordinates are $\left(x_{1},y_{1}\right)$; B is another imaging point for the width of the rectangular box: its coordinates are $\left(x_{2},y_{1}\right)$. $A^{\prime }$ and $B^{\prime }$ are the points opposite A and B on the road plane, respectively. The distance $d_{1}$ and $d_{2}$ between points $A^{\prime }$ and $B^{\prime }$ to the car camera can be calculated using Equation (1). It is easy to calculate the length w (the width of the road-surface pothole) of a rectangular body by using the principle of small-aperture imaging, which is derived as follows:(4)$$\cos\left(\alpha \right)=\frac{m_{1}^{2}+m_{2}^{2}-w^{2}}{m_{1}\times m_{2}}=\frac{l_{1}^{2}+l_{1}^{2}-m_{1}^{2}}{l_{1}\times l_{2}}$$

Among them,
$$\begin{array}{l} l_{1}=\sqrt{f^{2}+x_{1}^{2}+y_{1}^{2}}\\ l_{2}=\sqrt{f^{2}+x_{2}^{2}+y_{1}^{2}}\\ l_{3}=\mu \left(x_{2}-x_{1}\right)\\ m_{1}=\sqrt{d_{1}^{2}+h^{2}}\\ m_{2}=\sqrt{d_{2}^{2}+h^{2}} \end{array}$$

After substitution and simplification, the final expression of w is
(5)$$\begin{array}{l} w=\{2h^{2}+d_{1}^{2}+d_{2}^{2}-\frac{\left[2f^{2}+2y_{1}^{2}+x_{1}^{2}+x_{2}^{2}-\mu^{2}{\left(x_{2}-x_{1}\right)}^{2}\right]}{\sqrt{f^{2}+x_{1}^{2}+y_{1}^{2}}}{\times \frac{\sqrt{d_{1}^{2}+h^{2}}\times \sqrt{d_{2}^{2}+h^{2}}}{\sqrt{f^{2}+x_{2}^{2}+y_{1}^{2}}}\}}^{\frac{1}{2}} \end{array}$$

### 3.2. Length Calculation of Road-Surface Potholes

For the calculation of the width of road-surface potholes, as shown in Figure 6, let C be the first imaging point of the length of the rectangular frame outside the road-surface pothole, with coordinates $\left(x_{3},y_{3}\right)$; D is another imaging point for the length of the rectangular frame, with coordinates $\left(x_{3},y_{4}\right)$. $C^{\prime }$ and $D^{\prime }$ are the points opposite C and D on the road plane, respectively. The distance $d_{3}$ and $d_{4}$ between points $C^{\prime }$ and $D^{\prime }$ to the car camera can be calculated using Equation (1). It is easy to calculate the length $l_{hole}$ (the length of the road-surface pothole) of a rectangular body by using the principle of small-aperture imaging, which is derived as follows:(6)$$l_{hole}=d_{4}-d_{3}$$

Among them,
$$\begin{array}{l} d_{3}=\frac{h}{\tan\left(\partial +\arctan\left[\left(y_{3}-y_{0}\right)\mu /f\right]\right)}\\ d_{4}=\frac{h}{\tan\left(\partial +\arctan\left[\left(y_{4}-y_{0}\right)\mu /f\right]\right)} \end{array}$$

After substitution and simplification, the final expression of $l_{hole}$ is
(7)$$l_{hole}=\frac{h}{\tan\left(\partial +\arctan\left[\left(y_{4}-y_{0}\right)\mu /f\right]\right)}-\frac{h}{\tan\left(\partial +\arctan\left[\left(y_{3}-y_{0}\right)\mu /f\right]\right)}$$

### 3.3. Depth Calculation of Road-Surface Potholes

With the depth of the potholes having more influence on the vehicle driving than the width and length, how to accurately detect the depth of the potholes is the key of the proposed method in this paper. Based on the VIDAR principle, the feature points are marked, while the pothole depth is solved based on the feature point tracking using the images of the before and after two frames.

For the calculation of the depth of road-surface potholes as shown in Figure 7, let $E_{1}\left(x_{5},y_{5}\right)$ be the imaging point of the lowest point of the road-surface pothole on the $y_{1}$-axis in the previous image frame, and $E_{2}\left(x_{6},y_{6}\right)$ be the imaging point of the lowest point of the road-surface pothole on the $y_{2}$-axis in the image of the latter frame. $E^{\prime }$ is the point on the road plane corresponding to $E_{1}$ and $E_{2}$, $d_{5}$ is the distance between $E^{\prime }$ and the self-vehicle camera at the moment of the previous frame, $d_{6}$ is the distance between $E^{\prime }$ and the self-vehicle camera at the moment of the latter frame, Δd is the distance that the camera moves between the two frames before and after shooting (it is also the distance travelled by the vehicle during that time period), and $d_{5}=d_{6}+\Delta d$. Through small-hole imaging and the similarity principle of a triangle, the distance between the lowest point of the road-surface pothole and the self-vehicle camera can be calculated H (H=h+Δh). The specific derivation process is as follows:(8)$$\left\{\begin{array}{c} \frac{\mu \left(y_{5}-y_{0}\right)}{H}=\frac{f}{d_{5}}\\ \frac{\mu \left(y_{6}-y_{0}\right)}{H}=\frac{f}{d_{6}}\\ \begin{array}{c} \Delta d=d_{5}-d_{6}\\ H=h+\Delta h \end{array} \end{array}\right.$$

After substitution and simplification, the final expression of Δh is
(9)$$\Delta h=\frac{\mu^{2}\left(y_{5}-y_{0}\right)\left(y_{6}-y_{0}\right)\times \Delta d}{\mu \left(y_{6}-y_{5}\right)\times f}-h$$

### 3.4. Depth Update of Road-Surface Potholes

Due to the peculiarity of potholes, their depth cannot be measured accurately at the beginning like width and length, so this paper proposes a method to update the depth of road-surface potholes based on the pothole depth detection proposed in Section 3.3, the principle of which is shown in Figure 8. Where ① is the first image set, ② is the second image set and ③ is the third image set. Each image set contains two frames.

In the image frames, every two frames are divided into one group, and in the first group, VIDAR feature point marking is first performed through the first image frame, and the equation in (9) is used to solve the depth of the detected potholes in the first group $\Delta h_{1}$. The second and third groups repeat the process of the first group, and the corresponding detected depths $\Delta h_{2}$, $\Delta h_{3}$ respectively, are found; the maximum depth of the pothole is determined by comparing the size of $\Delta h_{1}$, $\Delta h_{2}$ and $\Delta h_{3}$.

### 3.5. VIDAR-Based Method for Pavement-Pothole Detection

1. Camera parameter updating based on IMU data

(1) Calibration of camera initial parameters. Calibrate the monocular camera mount-ed on a stationary vehicle, obtain the camera focal length f, mounting height h, and the pitch angle ∂, and obtain p (the pixel size of the photosensitive chip).

(2) Continuous inertial data acquisition. At the beginning of t=0, continuously acquire inertial data using IMU rigidly connected with monocular camera with frequency F.

(3) Camera parameter updating. Calculate Δd in period Δt according to inertial data.

2. Image processing and pothole detection

(1) Feature point extraction at t. Segment image at t, extract ROI, and extract points on the upper edge of ROIs as feature points.

(2) Horizontal length and width of the pothole calculation at t. Call camera parameters at t. Assuming feature points are located at the horizontal plane, calculate horizontal distance $d_{i}$ between feature points and the camera at t, and calculate the horizontal width w and length $l_{hole}$ of the pavement pothole at t.

(3) Feature point tracking and calculation of pothole depth. Acquire road image at $t\left(t=t+\Delta t\right)$, and extract the ROI and track the feature points at $t\left(t=t+\Delta t\right)$. Compare $H\left(H=h+\Delta h\right)$ and h (h is the height of the camera from the ground). If H≤h, the obstacle is a real obstacle (the obstacle has a certain height). If H≤h, the obstacle is a pothole. Calculate the depth of the pit Δh at t.

(4) Updated calculation of pavement pothole depth. Calculate the pavement pothole depth $\Delta h_{n}$ at time $t+\left(2n-1\right)\Delta t$. Compare the magnitude of $\Delta h_{n}$ and select the maximum value.

The flow chart of the VIDAR-based road-surface-pothole-detection method studied in this paper is shown in Figure 9.

## 4. Experiment and Analysis

The experiments in this paper included two parts: simulation experiments under controlled scenarios and real-vehicle experiments. According to the experimental results, it can be seen that the VIDAR-based road-surface-pothole-detection method proposed in this paper can effectively detect potholes in the road environment. Through comparison experiments with the classical method, our method was proved to have a high detection accuracy.

### 4.1. Simulation Experiments

On the experimental platform, experimental equipment such as IMUs and cameras were installed. To ensure the effectiveness of the experiment, a foam board with stickers was used as a road, while holes of different sizes were drilled in this foam board as road potholes. The video captured by the camera generated image sequences at a frame rate of 20 fps on which the road-surface-pothole-detection algorithm was experimented. In order to simulate the road environment more realistically, two types of environments were set up: a road environment with only one pothole, and a road environment with multiple potholes of different sizes, as shown in Figure 10.

The vehicles in the simulation experiment are car models and the potholes are man-made potholes. The length of the car model was 15 cm, the width was 5.8 cm, the wheel diameter was 2.9 cm, the wheel width is 0.8 cm, and the chassis height is 0.8 cm.

The feature region and feature point extraction process for road-surface potholes using the MSER-based fast image region-matching method is shown in Figure 11.

Figure 11 is divided into four stages. The blue asterisks in the second stage represent the MSERs feature points; the block colors in the third stage represent the obstacle contours identified through the feature points; and the yellow rectangular boxes in the fourth stage represent the obstacle frames extracted through the feature points.

After the potholes were successfully detected, the images were matched and feature points were removed from two consecutive frames using a fast MSER-based image-region matching method, as shown in Figure 12.

Figure 12a,b represent the processing for two consecutive image frames. The block shading represents the position of the obstacle in the second frame image. The green crosses represent the feature points of the first frame image, the red circles represent the corresponding feature points of the second frame image, and the yellow line segments indicate the correspondence.

The feature point matching results in Figure 12 revealed that, in addition to the studied road-surface pothole feature points, Figure 12a matched the vehicle’s edge feature points; Figure 12b matched the wall stain at the rear of the experimental platform. According to the road-surface-pothole determination process in Figure 9, the above redundant feature points do not affect the detection of road-surface potholes and the calculation of the length, width and depth of potholes.

The inspection accuracy described in this paper consists of two components: one is the ability to detect potholes; the other is the difference between the dimensional information of the detected potholes and the true measurement (i.e., measurement error). In particular, the ability to detect potholes is judged by the relevant parameters of the confusion matrix.

The accuracy (*A*), recall (*R*) and precision (*P*) of its ability to detect potholes were measured by four metrics *TP*, *FP*, *TN* and *FN*. Let *a* be a pothole correctly identified as a positive example, *b* be a pothole incorrectly identified as a positive example, *c* be a pothole correctly identified as a negative example, and *d* be a pothole incorrectly identified as a negative example. Then, $TP=\sum_{i=1}^{n}a_{i}$, $FP=\sum_{i=1}^{n}b_{i}$, $TN=\sum_{i=1}^{n}c_{i}$, and $FN=\sum_{i=1}^{n}d_{i}$.

Therefore, the accuracy (*A*), recall (*R*) and precision (*P*) are calculated, as shown in Equations (10)–(12):(10)$$A=\frac{TP+TN}{TP+TN+FP+FN}$$
(11)$$R=\frac{TP}{TP+FN}$$
(12)$$P=\frac{TP}{TP+FP}$$

At the same time, for the method proposed in this paper, the dimensional error in pit detection is calculated, as shown in Equation (13).
(13)$$M=\frac{\left(\frac{\left|w_{r}-w\right|}{w_{r}}\right)+\left(\frac{\left|l_{r}-l\right|}{l_{r}}\right)+\left(\frac{\left|p_{r}-p\right|}{p_{r}}\right)}{3}\times 100\%$$

Among them, M is the measurement error; $w_{r}$ is the actual width of the pothole; w is the measured width; $l_{r}$ is the actual length of the pothole; l is the measured length; $p_{r}$ is the actual depth of the pothole; and p is the measured depth.

The simulation experiment collected information from 15 potholes. The test results are shown in Table 1.

The data in Table 1 show that of the 15 sets of potholes, 13 can be correctly identified and 2 cannot be identified. Further analysis revealed that these 2 could not be identified because their pothole width and length were too short and were rejected by the algorithm in the image-matching process.

Through calculation, the proposed method in this paper was found to have an accuracy (*A*) of 86.67%, a recall (*R*) of 86.67% and a precision (*P*) of 100% in simulated experiments.

The difference between the dimensional information of the detected potholes and the true measurement is shown in Table 2.

In the analysis of the data in Table 2, the method proposed in this paper was found to have an average error of 4.76% in the detection of pit length, width and depth in simulated experiments.

Through analysis of the results in Table 1 and Table 2, it was concluded that the VIDAR-based road-surface-pothole-detection method proposed in this paper is effective in detecting potholes in simulated experiments.

### 4.2. Experiments on Real Vehicles

In the real-vehicle experiments, an electric vehicle was used as the experimental vehicle (as shown in Figure 13). The relevant equipment is as follows: the MV-VDF300SC industrial digital camera was mounted on the vehicle as a monocular vision sensor; the HEC295IMU was mounted on the bottom of the experimental vehicle for real-time positioning and reading of the vehicle’s movement; the GPS was used to pinpoint the vehicle’s position; and the computing unit was used for real-time data processing. The digital camera was a USB2.0 standard interface with the advantages of a high resolution, high accuracy and high definition, and the relevant parameters are shown in Table 3. In the actual calculation process, due to the complexity of multi-sensor data sources, fuzzy logic was used to deal with complex systems [21] to combine the acquired information in a coordinated way, to improve the efficiency of the system and to process the information efficiently in real scenarios.

In this paper, the Zhengyou Zhang calibration method was used to calibrate the camera. The calibration process and calibration procedure are shown in Figure 14.

The aberrations of the camera included three types of aberrations: radial aberrations, thin-lens aberrations and centrifugal aberrations. The superposition of by these three types of aberrations caused a nonlinear distortion, the model of which can be represented in the image coordinate system, as shown in (14).
(14)$$\left\{\frac{\delta_{x}\left(x,y\right)=s_{1}x\left(x^{2}+y^{2}\right)+2p_{1}xy+p_{2}y^{3}+k_{1}x\left(x^{2}+y^{2}\right)}{\delta_{y}\left(x,y\right)=s_{2}y\left(x^{2}+y^{2}\right)+2p_{2}xy+p_{1}y^{3}+k_{1}x\left(x^{2}+y^{2}\right)}\right.$$

Among them, $s_{1}$ and $s_{2}$ are the centrifugal aberration coefficients of the camera; $k_{1}$ and $k_{2}$ are the radial aberration coefficients of the camera; and $p_{1}$ and $p_{2}$ are the thin-lens aberration coefficients of the camera.

To facilitate the calculation, the centrifugal distortion of the camera is not considered in this paper, so the internal reference matrix of the camera used in this paper can be expressed as (15):(15)$$M=\left[\begin{array}{ccc} 5.9774\times 10^{3} & 0 & 0.9499\times 10^{3}\\ 0 & 5.9880\times 10^{3} & 0.3571\times 10^{3}\\ 0 & 0 & 1 \end{array}\right]$$

The calibration of the external parameters of the camera can be calculated by obtaining the edge object points of the lane line, and the calibration results are shown in Table 4.

Throughout the real-vehicle experiments, the IMU obtained the angular acceleration of the self-vehicle; the camera’s pitch angle was obtained and updated by solving the camera pose using the Quaternion method. The image was processed by a fast MSER-based image area-matching method, and the self-vehicle acceleration was used to calculate the horizontal distance between the self-vehicle and the road-surface pothole and the maximum width and maximum length of the road-surface pothole. The depth of the road-surface pothole was solved by using small-aperture imaging and a vertical distance from the obstacle that was always constant during real-vehicle movement.

Based on the above principles, the evaluation of the real-vehicle experiment was continued using the evaluation indicators in Section 4.1. Some of the test results are shown in Figure 15. The results of the experimental results are shown in Table 5 and Table 6.

Upon calculation, the proposed method in this paper had an accuracy (*A*) of 91.43%, a recall (*R*) of 91.43% and a precision (*P*) of 100% in the real-vehicle experiments.

The data in Table 5 show that of the 35 sets of potholes, 32 could be correctly identified and 3 could not be identified. Further analysis revealed that 3 sets of data could not be detected due to the depth of the pits being too shallow to be correctly identified as potholes.

In order to better cope with the different sizes and types of potholes in the road surface, we stipulated that one of the lengths or widths of the potholes used in the experiments should be greater than 150 mm, and the depth should be greater than 50 mm.

Analysis of the 35 sets of data shown in Table 6 revealed that the VIDAR-based road-surface-pothole-detection method proposed in this paper could detect the width, length and depth of road-surface potholes more accurately in the actual inspection process, and the average error for the 35 sets of data was 6.23%.

The experimental pits were categorized into six types based on length, width and depth, as shown in Table 7.

In order to further verify the accuracy and correctness of the method proposed in this paper, the detection results of the Faster-RCNN and YOLO-v5 detection methods were compared with the VIDAR-based pothole-detection method proposed in this paper.

Shaoqing Ren proposed the Faster-RCNN model in 2015 [22]. The model uses a small Region Proposal Network (RPN) instead of the Selective Search algorithm, which substantially reduces the number of proposal boxes, improves the drawback that Selective Search is too slow in generating proposal windows, and increases the processing speed of images.

YOLO (You Only Look Once) is one of the most typical algorithms in the field of target detection and is able to perform the task of target detection very well. The YOLO-v5, proposed by Glenn Jocher in 2020, introduces adaptive anchor-frame calculation and adaptive scaling techniques, which have the advantages of a simple structure, fast speed and high accuracy. For the YOLO-v5 [23] detection method, images of various types of potholes on the road were divided into a training set and a validation set, and the image sequences shown in Section 4.1 of this paper were used as the test set.

In the comparison experiment, the experimental vehicles were in the same position, stationary, and used the VIDAR-based pothole-detection method, the Faster-RCNN method, and the YOLO-v5 method, to detect 80 potholes of different sizes.

The results of the three methods are shown in Table 8, and the average error of the three methods in detecting the pothole information are shown in Table 9:

In conjunction with Table 8, the accuracy (*A*), recall (*R*), precision (*P*) and detection time of the three methods were further analyzed, as shown in Table 10.

Analysis of the experimental results in Table 9 and Table 10 reveals that the vidar-based pothole-detection method excludes the interference of road obstacles and only marks and calculates the non-road obstacle feature points, so it can detect potholes more correctly than faster-rcnn and yolo-v5s. from the overall results, we can see that as vidar solved the camera pose problem through imu, the vidar-based pothole-detection method proposed in this paper improved the correct rate by 16.89%, recall rate by 13.01%, accuracy by 6.04% and detection error by 12.45% compared to the faster-rcnn method; compared to the yolo-v5 method, the correct rate improved by 10.8%, the recall rate by 8.78%, the precision by 2.9% and the detection error by 8.4%. However, since the length, width and depth information of the pothole needs to be detected and the depth information of the pothole is obtained from the target tracking and data parsing of multiple images, the method proposed in this paper does not have a significant advantage in detection time.

## 5. Conclusions

This paper focused on detecting potholes and measuring their lengths, widths and depths by selecting and matching the feature points of the potholes and using the small-hole imaging principle of the camera. The research method in this paper can achieve the real-time detection of potholes on the road surface while the vehicle is moving, thus helping the vehicle to better avoid obstacles. This paper also focused on analyzing the methods for calculating the length, width and depth of potholes in the road surface, with particular emphasis on the method for updating the depth of potholes in the road surface based on feature point matching.

As described in Section 3 of this paper, relevant computational methods were analyzed for their length, width and depth information extraction of road-surface potholes, and depth-updating ideas and algorithms were provided for depth considering the field of view.

In Section 4, the VIDAR-based road-surface-pothole-detection method proposed in this paper was simulated in an indoor controlled scenario. The ability of the method to correctly detect potholes and the accuracy of the information related to the detection of potholes were evaluated through a unique evaluation system.

In addition, in Section 4, the VIDAR-based road-surface pothole algorithm proposed in this paper was verified to have a stronger detection capability and higher detection accuracy by comparing the experiments with the classical Faster-RCNN detection method and the YOLO-v5 detection method.

In summary, the VIDAR-based method for detecting potholes on roads proposed in this paper ensures that potholes can be identified in real time and accurately, and that the size and depth of the potholes can be detected during normal vehicle driving. This method avoids the delay of traditional road-maintenance work and has important research value for self-driving vehicles and active safety systems. The VIDAR detection method avoids the disadvantage of machine learning methods, which can only detect known types of obstacles. At the same time, machine learning cannot detect the exact size of the target obstacle, whereas the VIDAR detection method incorporates an IMU to obtain all information about the obstacle in front of the vehicle during travel by means of the camera imaging principle and pose analysis. The experimental results show that the method in this paper can effectively identify and detect potholes on the road surface and possess a stronger detection capability. Compared with the Faster-RCNN method, the correct rate was 16.89% higher, the recall rate was 13.01% higher, the accuracy was 6.04% higher and the detection error was 12.45% lower; compared with the YOLO-v5s method, the correct rate was 10.8% higher, the recall rate was 8.78% higher, the accuracy was 2.9% higher and the detection error was 8.4% lower, demonstrating that it can meet the requirements of detection in complex environments.

With the increase in vehicles on the road, the automatic detection of potholes on the road surface can effectively prevent road accidents, and at the same time can reduce the maintenance cost of the road. The VIDAR-based pothole-detection method integrates an IMU and a camera, can detect potholes of various sizes and types, and is a reliable and effective detection method.

However, due to the limitations of the experimental site, we were not able to provide more types of pavement pothole sizes and types; therefore, selected partial data from each type of pothole were examined. In addition, due to the failure to consider the vehicle’s own attitude, the road surface detected by the method in this paper must be flat and straight, and the size of the pothole must satisfy the rule that the length and width are both greater than 0.15 m. Also, due to the need to resolve the length, width and depth of the pothole, the method proposed in this paper does not have a significant advantage over traditional methods in terms of detection time. The next step can be to continue to develop the function of IMU in VIDAR and to explore the methods and strategies for pavement pothole detection in multiple scenarios.

## Figures and Tables

**Figure 1 sensors-23-07468-f001:**
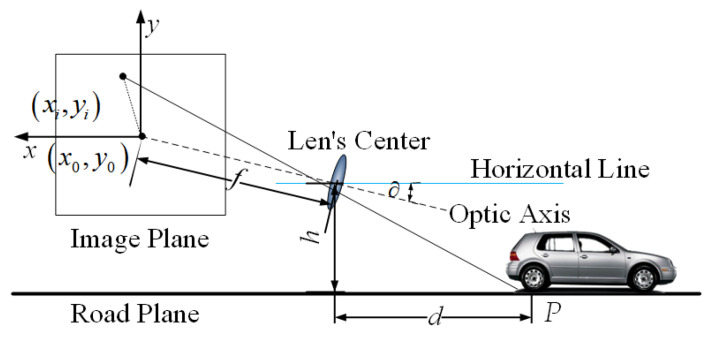
Three-dimensional obstacle small-hole imaging principle—diagram.

**Figure 2 sensors-23-07468-f002:**
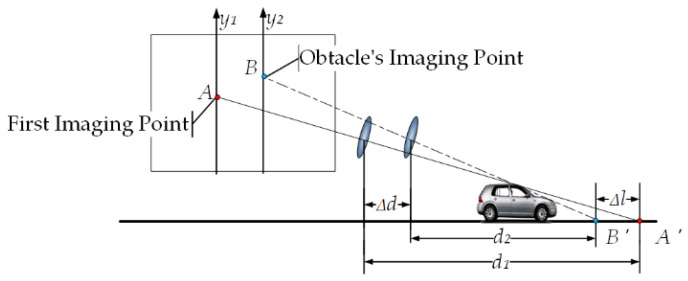
Static obstacle imaging schematic.

**Figure 3 sensors-23-07468-f003:**
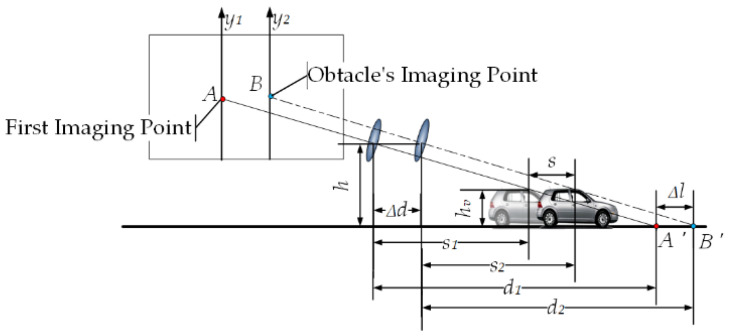
Dynamic obstacle imaging schematic.

**Figure 4 sensors-23-07468-f004:**
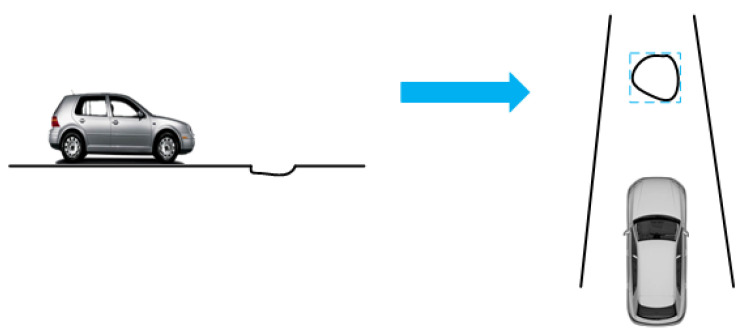
Road-surface pothole treatment diagram.

**Figure 5 sensors-23-07468-f005:**
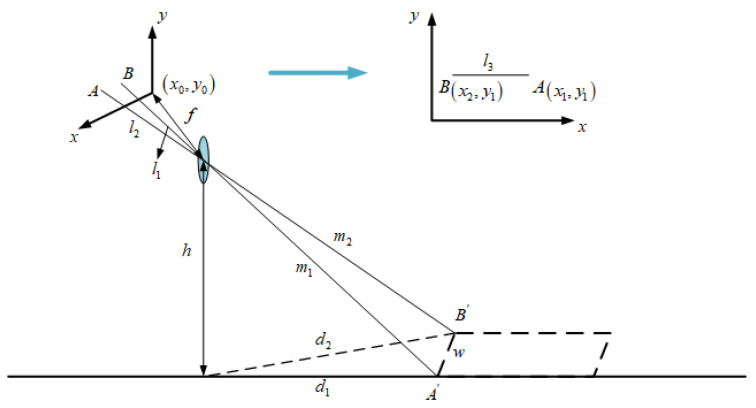
Road-surface-pothole width-calculation-method chart.

**Figure 6 sensors-23-07468-f006:**
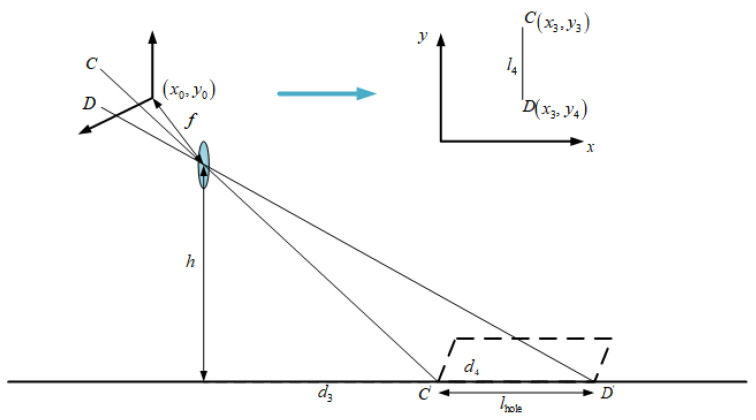
Roadway pothole length-calculation-method chart.

**Figure 7 sensors-23-07468-f007:**
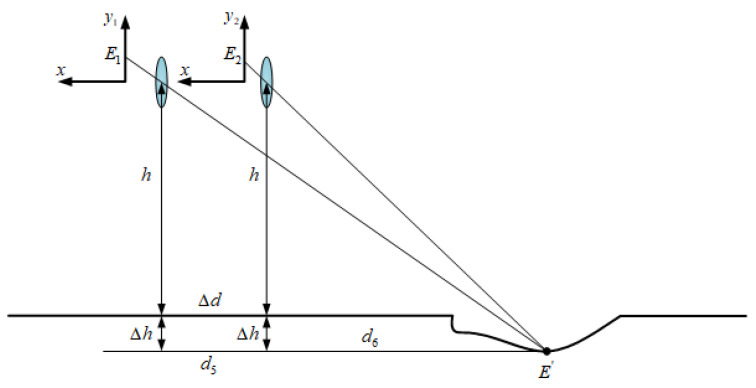
Road-surface pothole depth-calculation-method chart.

**Figure 8 sensors-23-07468-f008:**
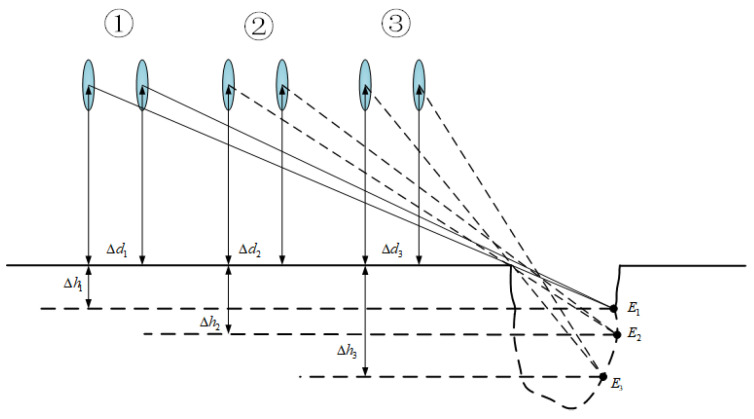
Schematic diagram of road-surface pothole depth update.

**Figure 9 sensors-23-07468-f009:**
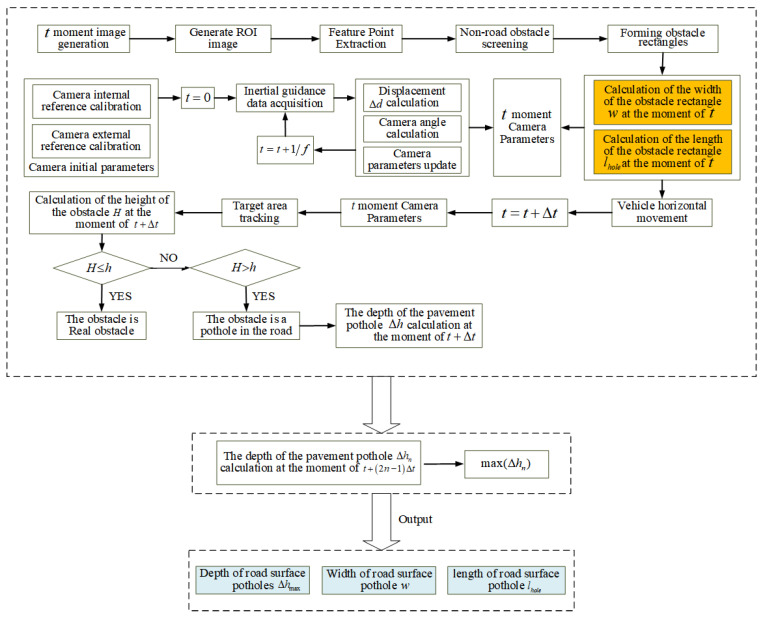
Flow chart of VIDAR-based road-surface pothole detection.

**Figure 10 sensors-23-07468-f010:**
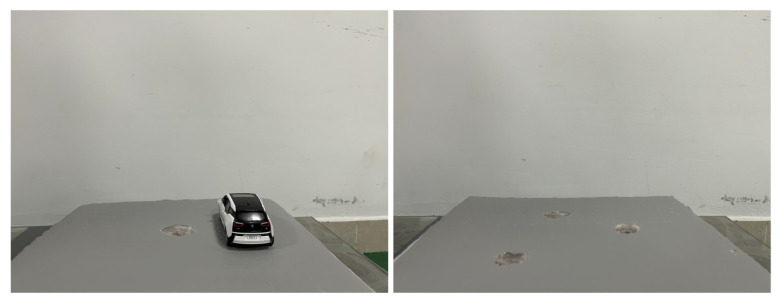
Schematic diagram of road-surface potholes under the simulation experiment.

**Figure 11 sensors-23-07468-f011:**
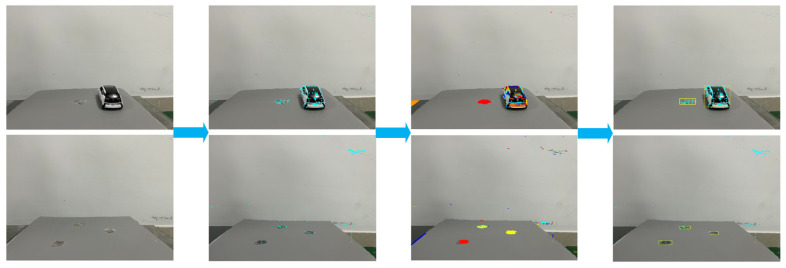
Feature point extraction and rectangular-box marking.

**Figure 12 sensors-23-07468-f012:**
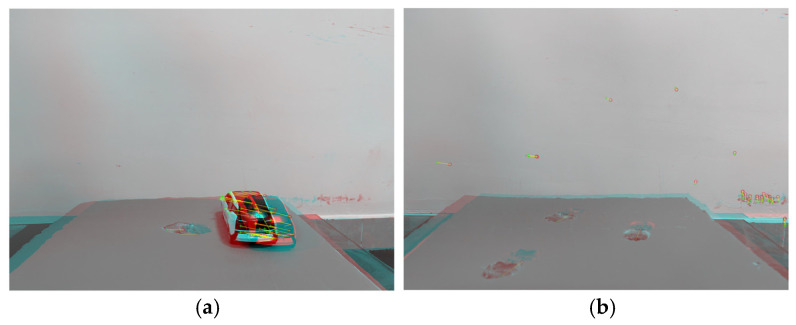
Feature point tracking and image matching.

**Figure 13 sensors-23-07468-f013:**
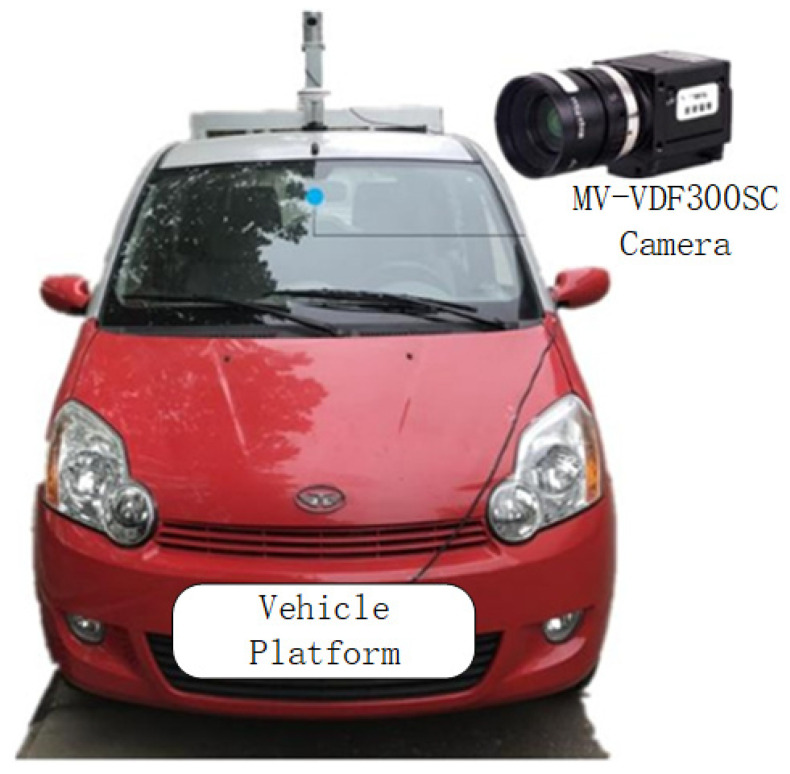
Experimental vehicle equipment diagram.

**Figure 14 sensors-23-07468-f014:**
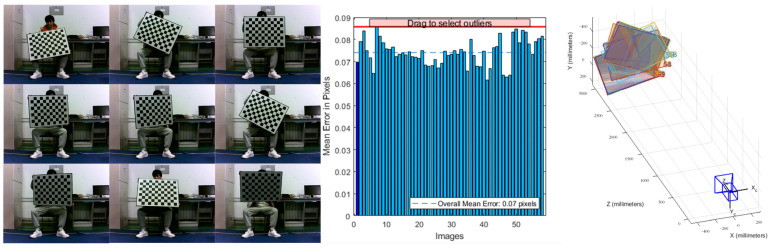
Camera calibration and alignment process.

**Figure 15 sensors-23-07468-f015:**
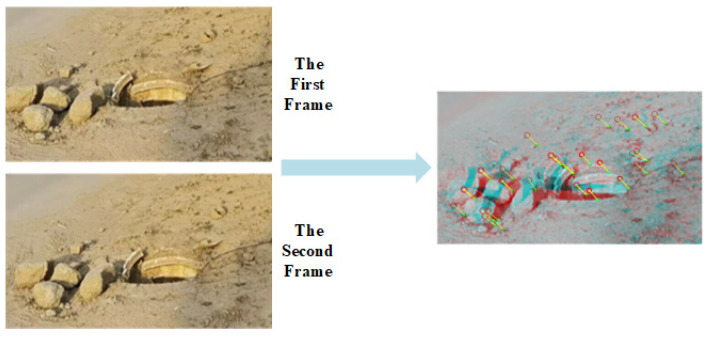
Selected results of pothole detection under the real environment.

**Table 1 sensors-23-07468-t001:** Simulation test results.

Evaluation Indicators	Numerical
*TP*	13
*FP*	0
*TN*	0
*FN*	2

**Table 2 sensors-23-07468-t002:** The difference between the pothole dimensional information detected in the simulation experiment and the real measurements.

Actual Width $w_{r}/cm$	Measured Width w/cm	Actual Length $l_{r}/cm$	Measured Length l/cm	Actual Depth $p_{r}/cm$	Measured Depth p/cm	Measured Errors /%
3.2	3.1	2.1	2.2	2.0	2.1	4.3
5.1	5.0	2.8	3.0	1.6	1.7	5.11
4.2	4.1	2.5	2.5	2.2	1.9	5.34
3.5	3.5	2.2	2.3	1.4	1.3	3.9
3.1	3.3	1.8	1.8	1.2	1.1	4.93
3.5	3.4	2.0	2.1	1.5	1.6	4.84
3.5	3.6	2.0	2.1	1.5	1.4	4.84
4.0	3.8	3.0	3.1	1.2	1.2	2.78
3.0	3.2	2.0	1.8	1.0	1.0	5.56
0.8	--	0.6	--	1.5	--	--
2.4	2.2	3.3	3.1	1.0	1.0	4.8
3.2	3.0	4.0	4.0	1.5	1.6	4.31
2.0	2.1	3.0	3.1	1.5	1.6	5
0.7	--	0.8	--	1.5	--	--
3.3	3.2	1.8	1.9	1.0	0.9	6.2

**Table 3 sensors-23-07468-t003:** Some performance parameters of the MV-VDF300SC camera.

MV-VDF300SC			
Highest resolution	2048 × 1536	Power requirements (V)	5
Output color	color	Power consumption (W)	Rated < 5
Frame rate (fps)	12	Operating temperature (°C)	0–60
Output method	USB2.0	Dimensions (mm)	43.3 × 29 × 29

**Table 4 sensors-23-07468-t004:** External parameter calibration of the MV-VDF300SC.

External Parameter Type	Parameter Size
Pitch angle	1.25
Yaw angle	1.65
Rotation angle	2.45

**Table 5 sensors-23-07468-t005:** Results of experimental tests on real vehicles.

Evaluation Indicators	Numerical
*TP*	32
*FP*	0
*TN*	0
*FN*	3

**Table 6 sensors-23-07468-t006:** The difference between the pothole dimensional information detected in the real network experiment and the real measurements.

Actual Width $w_{r}/m$	Measured Width w/m	Actual Length $l_{r}/m$	Measured Length l/m	Actual Depth $p_{r}/m$	Measured Depth p/m	Measured Errors /%
0.30	0.32	0.20	0.18	0.10	0.10	5.56
0.40	0.38	0.30	0.31	0.12	0.12	2.78
0.35	0.36	0.20	0.21	0.15	0.14	4.84
0.33	0.32	0.18	0.19	0.10	0.09	6.2
0.30	0.31	0.20	0.21	0.08	0.09	6.94
0.28	0.28	0.10	0.11	0.10	0.11	6.36
0.29	0.30	0.15	0.14	0.09	0.08	7.08
0.27	0.28	0.10	0.09	0.12	0.11	7.35
0.30	0.30	0.14	0.15	0.15	0.13	7.22
0.24	0.25	0.15	0.16	0.20	0.18	6.94
0.15	--	0.22	--	0.06	--	--
0.20	0.19	0.24	0.22	0.14	0.13	6.83
0.14	0.15	0.25	0.26	0.15	0.16	5.94
0.17	0.16	0.21	0.23	0.20	0.21	6.8
0.20	0.21	0.30	0.31	0.15	0.16	5
0.08	0.09	0.16	0.17	0.20	0.20	6.25
0.15	--	0.22	--	0.06	--	--
0.10	0.11	0.16	0.15	0.20	0.21	7.08
0.24	0.22	0.33	0.31	0.10	0.10	4.8
0.32	0.30	0.40	0.40	0.15	0.16	4.31
0.14	0.13	0.14	0.14	0.08	0.09	6.55
0.17	0.15	0.20	0.20	0.12	0.11	6.70
0.20	0.20	0.11	0.10	0.14	0.13	5.41
0.21	0.22	0.18	0.19	0.20	0.22	6.77
0.10	0.09	0.12	0.11	0.15	0.15	6.11
0.12	--	0.10	--	0.06	--	--
0.30	0.28	0.20	0.20	0.10	0.11	5.56
0.28	0.27	0.14	0.15	0.09	0.10	7.28
0.24	0.23	0.10	0.11	0.12	0.11	7.50
0.20	0.19	0.15	0.14	0.10	0.09	7.22
0.14	0.15	0.11	0.12	0.08	0.08	5.41
0.33	0.32	0.08	0.09	0.16	0.15	7.26
0.14	0.14	0.21	0.22	0.14	0.12	6.35
0.08	0.08	0.16	0.15	0.09	0.10	5.79
0.16	0.15	0.22	0.21	0.12	0.13	6.38

**Table 7 sensors-23-07468-t007:** Category of potholes.

Categories	Width	Length	Depth
$A_{1}$	≥0.15 m	≥0.15 m	≥0.06 m
$A_{2}$	≥0.15 m	≥0.15 m	≤0.06 m
$B_{1}$	≥0.15 m	≤0.15 m	≥0.06 m
$B_{2}$	≥0.15 m	≤0.15 m	≤0.06 m
$C_{1}$	≤0.15 m	≥0.15 m	≥0.06 m
$C_{2}$	≤0.15 m	≥0.15 m	≤0.06 m

**Table 8 sensors-23-07468-t008:** Results of the three methods.

Testing Method	*TP*	*FP*	*TN*	*FN*
VIDAR-based road-surface-pothole-detection Method	74	0	0	6
Faster-RCNN -based road-surface-pothole-detection Method	62	4	0	16
YOLO-v5s-based road-surface-pothole-Detection Method	67	2	0	13

**Table 9 sensors-23-07468-t009:** Comparison of the average error of three methods for detecting pothole information.

Testing Method	Average Error
vidar-based road-surface-pothole-detection method	6.86%
faster-rcnn -based road-surface-pothole-detection method	19.31%
yolo-v5s-based road-surface-pothole-detection method	15.26%

**Table 10 sensors-23-07468-t010:** Comparative accuracy of the three methods for the detection of the pothole information.

Testing Method	*A*	*R*	*P*	Time (*t*)
vidar-based road-surface-pothole-detection method	92.50%	92.50%	100%	0.096
faster-rcnn -based road-surface-pothole-detection method	75.61%	79.49%	93.94%	0.064
yolo-v5s-based road-surface-pothole-detection method	81.70%	83.75%	97.10%	0.035

## Data Availability

Not applicable.

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
