# Peer review of "VIDAR-Based Road-Surface-Pothole-Detection Method"

_sensors, 2023, doi:10.3390/s23177468_

Round 1
Reviewer 1 Report
1. The abstract should provide a high level overview of the overall content of the pothole detection study. It is recommended that the authors reorganise the illumination by summarising the research background, core methods and evaluation indicators.
2. It is recommended that the authors add representative references and summarise current research difficulties or research gaps in the introduction section.
3. It is suggested that the authors summarise the innovative nature of this paper at the end of the introduction.
4. Is the "3.2.1 An Obstacle Region Extraction Method Based on MSER Fast Image Matching" under the heading of 2.1 correct? Please check.
5. The schematic diagrams in the text are not clear. The author should explain each schematic diagram in detail.
6. The simulated data corresponds to the principle section, but does the urban and rural perspective in the real scenario section of 4.2 correspond directly to the principle section? If this has changed please ask the author to elaborate.
7、Why was YOLOV5 used as a comparison method and what was the basis for the choice? The mere use of yolov5 as a comparison does not prove the superiority of the proposed method, and it is suggested that the authors add more comparative experiments of the updated method.
8. The English writing is poor. The text contains incomplete words for individual headings, improper use of individual vocabulary, and unclear expressions for individual statements. The authors are requested to double-check.
The English writing is poor. The text contains incomplete words for individual headings, improper use of individual vocabulary, and unclear expressions for individual statements. The authors are requested to double-check.
Author Response
Thank you for your letter and reviewers' comments on our manuscript entitled "VIDAR-based road surface pothole detection method (Manuscript-ID:sensors-2479252)". These comments were very valuable and helped us to revise and improve the paper. We have carefully studied all the comments and have made careful revisions. We have made some changes to the English grammar and vocabulary in the manuscript.The major revisions and responses to the reviewers' comments are marked in the paper. Attached are the complete revised manuscript and the manuscript with revision marks, respectively. Below are our responses to all reviewers' comments.

Reviewer 2 Report
The main contribution of this paper is that the authors propose an algorithm for detecting potholes through VIDAR using the MESR-based fast image region matching method, and here are some suggestions in my view to improve the quality of this paper:
1、 There are many grammatical errors in the manuscript, such as "which can classified as", “inseetionresnetv2”, and “can be effectvely detected”. Also, the typography of this manuscript is also poor. The authors need to carefully check and revise the manuscript.
2、 The introduction section lacks a comprehensive summary of the current research results on pothole detection and and requires the addition of an overview of the latest works.
3、 Chinese text appears in Figure 13.
4、 The ViDAR device in the experiment is based on IMU and camera, but the manuscript lacks a description of IMU data processing and how the IMU and camera cooperate.
5、 The simulated experiment contains insufficient data, and lacking persuasiveness. In fact, the academic community has open-sourced numerous datasets for pothole detection, such as the dataset mentioned in the paper Pothole Detection Based on Disparity Transformation and Road Surface Modeling, which can be downloaded from https://github.com/ruirangerfan/stereo_pothole_datasets. It is recommended that the authors utilize open-source datasets for comparison in the paper.
6、 The manuscript completely lacks any description of experimental scenarios in real environments, leaving readers unable to understand the testing conditions and the number of scenarios. It is strongly recommended to include relevant content on real environment data and show partial pothole detection results in the manuscript.
7、 The author only compared YOLOv5 algorithm with the proposed algorithm. If possible, it would be advisable to add comparisons with more recent algorithms.
The writing demonstrates significant grammatical errors, making it difficult to comprehend the intended message.
Author Response

(The authors gave the same response as above.)

Reviewer 3 Report
General comments:
1. The current scientific literature addressed the problem of pothole or other road distresses detection. Authors may add and discuss more current literature addressed this subject.
The novelty of presented approach in comparison to the current state should be outlined.
References:
Ragnoli, Antonella, Maria Rosaria De Blasiis, and Alessandro Di Benedetto. "Pavement distress detection methods: A review." Infrastructures 3.4 (2018): 58.
Ma, N., Fan, J., Wang, W., Wu, J., Jiang, Y., Xie, L., & Fan, R. (2022). Computer vision for road imaging and pothole detection: a state-of-the-art review of systems and algorithms. Transportation safety and Environment, 4(4), tdac026.
Kim, Y. M., Kim, Y. G., Son, S. Y., Lim, S. Y., Choi, B. Y., & Choi, D. H. (2022). Review of Recent Automated Pothole-Detection Methods. Applied Sciences, 12(11), 5320.
HEDENSTRÖM, Linus; ERIKSSON, Sebastian. An investigation of detecting potholes with UAV LiDAR and UAV Photogrammetry. 2021.
Zhou, Y., Guo, X., Hou, F., & Wu, J. (2022). Review of intelligent road defects detection technology. Sustainability, 14(10), 6306.
2. Authors should discuss the dimensions of potholes (road distresses) of real network and ability of the presented VIDAR approach to detect them.
3. Discuss the benefit (time, finance) of presented method in comparison to current road distress scanning systems by road network administrations.
Specific comments:
1. Page 3
Shortages MSER, MESR are not introduced at the first occurrence.
2. Figs. 2, 3, 5, 6, etc. should enlarge. The readability of the images is poor.
Page 7 - Fig. 6
COMMENT: Where are points C, D, C’, D’ in Fig. 6?
Page 11
“Therefore, the accuracy (A), recall (R) and precision (P) are calculated as shown in equation (10)”
COMMENT: In Eqs (10)-(12)
Page 9, Section 4.1
COMMENT: Add the wheel dimension in Section 4.1 in relation to the pothole dimension. It is real vehicle or model of smaller dimensions?
Discuss the correlation of the potholes dimensions in Table 2 with dimensions of real road distress. How is calculated the measured errors in last column for three listed dimensions of pothole?
Page 15
COMEMNT: Add the reference to “YOLO-v5 detection method”. Who proposed and used this method for what purposes is the method intended?
Page 16 – Table 9
COMMENT: Authors stated that Bhatia [7] detected the potholes with higher accuracy. The authors should add further references and obtained precision of other methods.
Author Response

(The authors gave the same response as above.)

Round 2
Reviewer 1 Report
None
Minor editing of English language required
Author Response

(The authors gave the same response as above.)

Reviewer 2 Report
-
Please give more experiments to
show detection effectiveness for different types of defects.
not bad
Author Response
Thank you for your letter and reviewer's comments on our manuscript entitled "VIDAR-Based Method for Pothole Detection in Road Surfaces (Manuscript-ID: sensors-2479252)". These comments are extremely valuable and help us revise and improve the document. We have carefully studied all reviews and made careful revisions. Major revisions and responses to reviewer comments are indicated in the paper. Attached is the full revised manuscript and the manuscript with revision marks. Below are our responses to all reviewer comments.

Reviewer 3 Report
The quality of paper was partially improved. Several comments should be addressed.
General comments
1. Authors did not respond on the comment “Authors should discuss the dimensions of potholes (road distresses) of real network and ability of the presented VIDAR approach to detect them.”
Authors only added Fig. 14 without any explanation. It is real pothole or some artificial sample?
Authors showed in Table 2 that two potholes of the depth 0.06 m were not detected. These two potholes are of the lowest depth from the 20 processed potholes. Are such potholes of the depth 0.06 m and lower typical or marginal at current road network?
The comparison with real network distresses dimension would be welcome. Authors totally ignored this comment.
Miller and Bellinger (2014) in “Distress Identification Manual for The Long-Term Pavement Performance Program (Fifth Revised Edition)”
https://www.fhwa.dot.gov/publications/research/infrastructure/pavements/ltpp/13092/13092.pdf
specified the distress „Potholes“ as follows
POTHOLES
Description
Bowl-shaped holes of various sizes in the pavement surface. Minimum plan dimension is 150 mm. Circular potholes should have a minimum diameter of 150 mm. A 150-mm-diameter circle should fit inside irregular-shaped potholes.
Severity Levels : LOW < 25 mm deep, MODERATE 25 to 50 mm deep, HIGH > 50 mm deep
--
Thus the accuracy of the proposed method should be related to the real road distresses and its dimension.
2. Benefits or limits of the approach in comparison to current proposed or used approaches should be added in the paper. This was not responded.
3. Authors stated in response to the Comment #8 “the vehicles in section 4.1 are smaller sized car models.”
COMMENT: I have not found the information about car model and its size in the revised paper. This should be explained in the text.
4. The authors introduced the equation (13) for measurement error.
After control with data in Table 2 it seems that this new equation is illogical and did not match the presented values.
M = 100*(abs(3.2-3.1) + abs(2.1 - 2.2) + abs(2-2.1))/3 = 10 %
Table 2 stated the value 4.3 %.
It seems that the measurement error was calculated as
M = 100*( abs(w - wr)/wr + abs(l - lr)/lr + abs(p - pr)/pr)/3 = 4.3 %
Add an explanation.
Specific comments
Page 15,
“Section 4.1. Some of the test results are shown in Figure 15.”
COMMENT: In the manuscript only Fig. 14 is presented.
Page 16
The reference (literature e source) of the method yolo5 was still not added.
Page 16 – Table 6
COMMENT: Are the used 20 potholes real or artificial? Please explain.
Page 17
“and 80 potholes of the same position and size“
COMMENT: This statement is not clear. Authors used 20 sets of potholes. List the dimensions and position of 80 potholes.
Author Response

(The authors gave the same response as above.)
